# BET Bromodomain Degradation Disrupts Function but Not 3D Formation of RNA Pol2 Clusters

**DOI:** 10.3390/ph16020199

**Published:** 2023-01-29

**Authors:** Diana H. Chin, Issra Osman, Jadon Porch, Hyunmin Kim, Kristen K. Buck, Javier Rodriguez, Bianca Carapia, Deborah Yan, Stela B. Moura, Jantzen Sperry, Jonathan Nakashima, Kasey Altman, Delsee Altman, Berkley E. Gryder

**Affiliations:** 1Department of Genetics and Genome Sciences, Case Western Reserve University School of Medicine, Cleveland, OH 44106, USA; 2Certis Oncology Solutions, San Diego, CA 92121, USA; 3Kasey Altman Research Fund, Rein in Sarcoma, Fridley, MN 55432, USA; 4Case Comprehensive Cancer Center, Case Western Reserve University, Cleveland, OH 44106, USA

**Keywords:** BET bromodomains, BRD4, PAX3-FOXO1, Rhabdomyosarcoma, RNA Polymerase II, transcription, core regulatory circuitry

## Abstract

Fusion-positive rhabdomyosarcoma (FP-RMS) is driven by a translocation that creates the chimeric transcription factor PAX3-FOXO1 (P3F), which assembles de novo super enhancers to drive high levels of transcription of other core regulatory transcription factors (CRTFs). P3F recruits co-regulatory factors to super enhancers such as BRD4, which recognizes acetylated lysines via BET bromodomains. In this study, we demonstrate that inhibition or degradation of BRD4 leads to global decreases in transcription, and selective downregulation of CRTFs. We also show that the BRD4 degrader ARV-771 halts transcription while preserving RNA Polymerase II (Pol2) loops between super enhancers and their target genes, and causes the removal of Pol2 only past the transcriptional end site of CRTF genes, suggesting a novel effect of BRD4 on Pol2 looping. We finally test the most potent molecule, inhibitor BMS-986158, in an orthotopic PDX mouse model of FP-RMS with additional high-risk mutations, and find that it is well tolerated in vivo and leads to an average decrease in tumor size. This effort represents a partnership with an FP-RMS patient and family advocates to make preclinical data rapidly accessible to the family, and to generate data to inform future patients who develop this disease.

## 1. Introduction

Fusion transcription factors drive many types of human cancer [1]. Rhabdomyosarcoma is a cancer of the skeletal muscle lineage that affects children and young adults and has an overall five-year mortality rate of around 25% [2,3]. Prognosis is further affected by molecular subtype, other mutations, and age, with urgent treatments needed that address subtypes with the worst prognoses [2,4]. The fusion-positive subtype (FP-RMS) confers a less favorable prognosis and is defined by the translocation t(2;13)(q35;q14) that leads to a PAX3-FOXO1 transcription factor fusion (P3F) [5,6].

We have shown previously that P3F hijacks the process of myogenic differentiation by inducing de novo super enhancers, or clusters of highly active enhancers, at oncogenes and transcription factors (TFs), including master regulators MYOD, MYOG, SOX8, MYCN [7], and FOXF1 (ref. [8]). P3F also recruits co-activator proteins such as p300, BRD4, and MED1 and creates super clusters, or large multi-protein aggregates acting together to drive transcription at super enhancers [7,9]. Active histone marks are deposited by p300 and recognized by BRD4, which facilitates transcription elongation. The super clusters assembled at myogenic TFs such as MYOD1 maintain expression and a myoblast-like state, and PAX3-FOXO1 is locked by an infinite loop of myogenic enhancer logic via the myogenic TF-responsive super enhancer on the FOXO1 side of the translocation [10]. Derailed myogenic cell fate in RMS also corresponds with aberrant expression of myogenic ncRNAs [11], and PAX3-FOXO1 specifically drives miRNAs that contribute to disease pathogenesis [12]. The transcriptional repression of correct myogenic programs is also coordinated by epigenetic machinery, such as the NCOR/HDAC3 complex [13], the CHD4/NuRD complex [14], and the BAF complex [15]. These studies collectively illustrate that targeting chromatin-associated factors may be the key to therapeutically intervening for patients with FP-RMS.

BET bromodomains recognize histone acetylation, and the family of proteins that contain them (BRD4 especially) are essential to the function of histone acetylation-rich super enhancers [16]. It has also previously been demonstrated that JQ1, a BRD4 inhibitor, decreased proliferation of FP-RMS models in vitro and in vivo [7,17]. We sought to test it alongside newer BRD4-targeting molecules in several FP-RMS models, including those with high-risk mutations.

JQ1 is a thienotriazolodiazepine and a first-in-class, selective BET inhibitor that most strongly binds to BRD4 compared to other bromodomain proteins and can displace BRD4 from chromatin [18]. However, it has a relatively short half-life of one hour after IV administration of 5 mg/kg in mice [18]. A structurally similar BET inhibitor with improved pharmacokinetics, OTX015, showed antitumor effects in vitro and reached clinical trial, but showed toxicities that required intermittent dosing schedules [19,20]. This issue of toxicity in clinical trials has plagued other BET inhibitors, such as BAY 1238097 and GSK525762, which could be due to a small therapeutic window or lack of BET bromodomain selectivity [21,22].

In the present study, we test ARV-771, is a small-molecule proteolysis-targeting chimera (PROTAC) that links BET bromodomain proteins to an E3 ubiquitin ligase von Hippel Lindau, triggering ubiquitination and proteasomal degradation [23]. The BET-targeting arm of ARV-771 has a similar structure to JQ1 (ref. [23]). We also test BMS-986156, a BET bromodomain inhibitor with a carboline-based structure that showed favorable pharmacokinetics and anticancer activity against many different PDX models [24]. An advantage of degraders over inhibitors is that inhibitors must remain bound to exert their effect, while a single degrader molecule is released once degradation is complete, allowing for molecular recycling and potential gains in potency, and safety.

We test these molecules in fusion-positive models with additional mutations, including in *TP53*, which has been suggested to be particularly aggressive if present upon diagnosis [4], as well as *MYCN* amplification, which has also been correlated with worse outcomes [25], to expand the preclinical data available for different types of FP-RMS.

## 2. Results

### 2.1. BET Bromodomain Inhibition and Degradation Halt Cell Growth and Core Regulatory Circuitry in Rhabdomyosarcoma

To confirm the degradation of BRD4, we treated the FP-RMS cell line RH4 with BRD4 degrader ARV-771 and imaged after six hours. The BRD4 signal localized to the nucleus, as expected with the vehicle alone (DMSO), but the signal was lost after ARV-771 treatment, suggesting that the protein was being effectively proteolyzed (Figure 1a).

Next, we assessed the impact of BRD4 inhibition and degradation on cell proliferation. The FP-RMS cell line RH41 was treated with the BRD4 degrader ARV-771 or the inhibitors JQ1 or BMS-986158. JQ1 had an IC_50_ over 5 µM, while ARV771 had an IC_50_ of 297.7 nM, showing the same efficacy at a roughly 17-fold lower concentration for the degrader. BMS-986158 had an IC_50_ concentration of 9.507 nM (Figure 1b). Comparing the effects of JQ1 and ARV-771, which have similar BET-binding moieties, suggests that degradation is more potent than inhibition for a given BRD4-binding moiety, while the superior potency of BMS-986158 compared to ARV-771 shows that inhibition can outperform degradation with an improved BRD4-binding moiety.

To compare the effect of bromodomain inhibition versus degradation on gene expression, we conducted RNA-seq on ARV-771- and JQ1-treated RH4 and RH5 cells. RH5 is also FP-RMS with *MYCN* amplification and mutant p53. We used spike-in to normalize total reads based on cell number and found global decreases in transcription with ARV-771, but not JQ1 treatment (Appendix A). We used principal component analysis to differentiate expression profiles, finding that samples clustered as expected based on cell line identity along one axis, while the non-treated samples, DMSO, and JQ1 conditions formed a separate cluster from the ARV-771-treated cells along the second principal component axis (Figure 1c). The difference in transcription after ARV-771 treatment that was large enough in magnitude to cause JQ1 to cluster near the controls.

Chemical genomics previously taught us that inhibition at any node along the acetylation axis caused selective downregulation of core regulatory circuitry, including BRD4 inhibition [26]. Therefore, here we evaluated the expression of core regulatory TFs of FP-RMS and housekeeping genes. ARV-771 more aggressively downregulated core regulatory TFs compared to JQ1 (Figure 1d and Appendix A). Of the core regulatory TFs, two in particular stood out: *MYOD1* and *MYCN*. These genes showed dramatically decreased TPM levels after 6 h of treatment with ARV771 compared to non-treated and vehicle conditions as well as JQ1 (Figure 1e and Appendix A). ChIP-seq from RH4 cells alongside RNA-seq tracts from RH4 and RH5 at the *MYOD1* locus showed P3F co-binding with H3K27ac, BRD4, and Pol2 in upstream enhancers looped to the *MYOD1* gene by Pol2 HiChIP (Figure 1f). The RNA-seq tracts reiterated that ARV-771 treatment, and to a lesser extent JQ1 treatment, led to a striking decrease in the *MYOD1* transcript. GSEA analysis revealed that all PAX3-FOXO1 target genes and FP-RMS core regulatory TFs were negatively enriched with both treatments (Figure 1g).

### 2.2. RNA Polymerase II Loops Increase after Degradation of BET Bromodomains

To further investigate the effects of the loss of BET bromodomain function on transcription, we mapped RNA Pol2 contacts after BET bromodomain inhibition or degradation via AQuA-HiChIP [27] of Pol2. At baseline, mapping Pol2 contacts in RH4 revealed loops as expected at previously annotated super enhancers such as those upstream of *MYOD1* (Figure 2a). Interestingly, BET bromodomain degradation via ARV-771 led to an increase in total Pol2 loops compared to DMSO treatment (Figure 2b).

To examine the localization of Pol2 that might underly the increase in looping, we also ran our HiChIP data through a ChIP-seq pipeline to visualize Pol2 binding and overlaid with this with BRD4 ChIP-seq. The distribution of Pol2 at the MYOD1 locus showed that the effect of BRD4 degradation on Pol2 binding was not uniformly distributed. Quantifying the difference between DMSO and ARV-771 reads at this locus revealed an increase in Pol2 in the enhancers, the promoter, and the gene body, with a decrease after the transcriptional end site (Figure 2c). The distribution of Pol2 was quantified by introducing a new metric inspired by our observations: the loading ratio, defined as the ratio of Pol2 signal in the promoter and gene body divided by the amount of Pol2 after the transcriptional end site. Because of the known role of BRD4 in the control of pause–release and elongation, we also calculated the pause ratio, the amount of Pol2 between −30 bp and +300 bp relative to the transcriptional start site at the early elongation checkpoint, compared to the amount to actively elongating Pol2 in the body of the gene (Figure 2d). At core regulatory TF genes such as *MYOD1* and *MYCN*, the average loading ratio increased with ARV-771 treatment, while the pause ratio stayed the same or was slightly decreased. Unlike CRTF genes, for most P3F target genes, there was an increase in the pausing ratio (also known as the traveling ratio) with both BRD4 inhibition via JQ1 and degradation via ARV-771 (Appendix A), consistent with the known role of BRD4 in facilitating pause–release. This unique control of transcription and Pol2 dynamics by BRD4 at CRTF genes is unexpected, and certainly provokes new questions about the nature of BRD4 function in the extended multi-super-enhancer clusters in which CRTFs are regulated.

### 2.3. BET Bromodomain Inhibitor BMS-986158 in an FP-RMS In Vivo Model

Based on the promising in vitro results and the importance of preclinical data for FP-RMS patients deciding on clinical trial enrollment, we moved toward testing BET bromodomain-targeting drugs in vivo in an O-PDX mouse model of FP-RMS and CRT00513 (P3F+, *MYCN* amplification, mutant p53). The treatment with BMS-986158 was well tolerated, with all mice surviving to the study end point. The average tumor volume decreased over time with BMS-986158 treatment, while the average tumor volume increased with vehicle alone, although the results were not statistically significant (Figure 3a). Looking in more detail at the tumor volumes in individual BMS-986158-treated mice revealed four responders and three non-responders out of seven altogether (Figure 3b). Among the four responders, one showed an effect by day 6, while the other three responded around day 23–27. For the three non-responders, one tumor volume stayed the same, while the other two tumors grew. For the vehicle-treated mice, there was also some variation, with only three out of seven showing increases in tumor volume at each time point, although all had higher volumes on day 27 compared to day 2. Tumor volumes were estimated via MRI (example images in Figure 3c). To assess the impact of the drug on PAX3-FOXO1 target genes in vivo, we dosed three mice for 3 days with BMS-986158 or vehicle for 3 days, extracted tumors, and performed RNA-seq. GSEA revealed a clear and selective downregulation of PAX3-FOXO1 target genes (Figure 3d), indicating on-target activity in vivo. However, despite the global trends, zooming in on individual PAX3-FOXO1 target genes revealed variability of response (Figure 3e), mirroring the variability of tumor growth reduction in the BMS-986158-treated PDX tumors.

We also evaluated BMS-986158 via RNA-seq analysis in both *MYCN*-amplified and non-*MYCN*-amplified cells. GSEA showed that FP-RMS core regulatory TFs were negatively enriched with BMS-986158 treatment in the P3F+ cell lines RH41 and RH5, regardless of *MYCN* amplification status (Appendix A).

## 3. Discussion

The comparison between structurally similar BET bromodomain inhibitors and degraders such as JQ1 and ARV-771 suggests that for a given inhibitor, tethering to a ligand for E3 ubiquitin ligase can create a much more potent PROTAC. The high performance of the inhibitor with a different structure, BMS-986158, relative even to the degrader ARV-771, suggests that converting this small molecule to a PROTAC degrader could yield the most potent effect. The in vivo study also showed that BMS-986158 treatment was well tolerated, with no study animals showing excessive weight loss before the end point. This is important because of safety and tolerability concerns with previous BETi clinical trials.

In terms of efficacy, the tumors in BMS-986158-treated animals decreased in size on average, but the results show a large degree of heterogeneity. Larger studies have demonstrated high variability of response in groups of mice bearing the same PDX model and experiencing the same treatment [28]. Future pharmacokinetic and pharmacodynamic studies may also provide further insight into the variation in response. Differences in tumor penetration are a major source of variation in PDX models, such as has been directly measured by others [29]. There was high variability in three canonical PAX3-FOXO1 target genes, *PIPOX*, *FGFR4*, and *ASS1*, in three mice treated with BMS-986158 (Figure 3e). As these genes are markers of BRD4 inhibition in this disease [7], we infer that these genes reflect differences in drug uptake in the tumors, a likely cause of variability in tumor growth dynamics in the group of animals treated with BMS-986158.

This study also raised interesting biological questions about RNA Pol2. BET bromodomain degradation by ARV-771 drastically reduced transcription, but there was a counterintuitive increase in RNA Pol2 looping. These loops primarily involved enhancers connected to the TSS and gene body. One theoretical explanation is that ARV-771 could increase RNA Pol2 looping by stabilizing droplet formation, but this contradicts prior findings that BRD4 inhibition by JQ1 dissolves RNA Pol2 clusters, although degradation could have a different effect [30]. It has also been shown that BRD4 is essential for CDK9 recruitment to Pol2, and phosphorylation of serine 2 in the RNA Pol2 C-terminal domain by CDK9 drives the transition from pausing to elongation [31]. The increase in the RNA Pol2 loading ratio, but not the pausing ratio, suggests a separate mechanism from CDK9. CDK7 phosphorylates serine 5 of Pol2 at the promoter to promote initiation [32], and it has been shown that CDK7 inhibitors have a synergistic effect when combined with BRD4 inhibitors in neuroblastoma [33], so a local increase in CDK7 could also be implicated. Another possibility is that BRD4 at super enhancers is involved in the unloading of Pol2 during termination. Future studies of specific CDK inhibition followed by measurement of nascent RNA in the context of BET degradation will help to clarify this question.

Much of this work, particularly the in vivo study, was enabled by patient and family advocates seeking to inform clinical decision making for the wider FP-RMS community. The in vivo study involved a partnership between teams and institutions to create a representative PDX model for one patient with relapsed FP-RMS with mutant *TP53* and amplified *MYCN* following conventional treatment. While generating PDXs may not be a scalable endeavor, making preclinical data from diverse models intuitive and widely available to patients and care teams is critically important to inform treatment decisions after disease recurrence.

## 4. Methods

### 4.1. Chemicals

JQ1 was a generous gift from Jun Qi (Dana-Farber Cancer Institute, Boston, MA, USA). ARV-771 and BMS-986158 were obtained from Selleck Chem (#S8532, #S9691). For in vitro studies, compounds were resuspended in DMSO. For the in vivo study, BMS-986158 was formulated by adding 10% DMSO, 40% PEG300, 5% Tween-80, and 45% saline, in that order.

### 4.2. Tissue Culture

FP-RMS cell lines RH4 and RH5 (from Javed Khan) and RH41 (from Aru Narendran) were cultured in DMEM (Gibco #10313039) supplemented with 10% FBS (Gibco #A3160402, Waltham, MA, USA) and 1x penicillin–streptomycin–glutamine (Gibco #10378016). Mouse myoblast C2C12 cells from (ATCC #CRL-1772) were grown in the same medium to generate lysates for HiChIP spike-in.

### 4.3. Cell Proliferation

To determine IC_50_, RH41 and RH5 cells were plated at 2000 cells/well into 384-well plates (Corning #2764) and treated the next day with a range of drug doses (0.2 nM–30 µM). Cell confluency was then monitored in an Incucyte Zoom and imaged every 6 h until DMSO controls became confluent. IC_50_ values were calculated with GraphPad PRISM software.

### 4.4. Immunocytochemistry

RH4-BRD4-mGFP were seeded at 85,000 per well into chambered cover glass (ThermoFisher #155379PK, Waltham, MA, USA). The following day, they were treated with 100 nM ARV-771 or an equal volume of DMSO (0.1%). Then, 6 hours later, they were fixed with 1 mL of 4% formaldehyde for 14 min at RT, and quenched with 200 µL of glycine for 5 min at RT. They were subsequently washed with PBS 3x, permeabilized with 0.5% Triton X-100 in TE pH 7.4 for 15 min at RT, washed with PBS 3x, and blocked with 4% FBS in PBS-T (PBS + 0.1% Triton X-100) for 1 h at RT. The following primary antibodies and dilutions were then applied and incubated overnight at 4 °C: rabbit anti-BRD4 (Abcam #ab128874, Cambridge, UK) 1:200, mouse anti-MyoD (Invitrogen #MA1-41017, Waltham, MA, USA) 1:200 in PBS-T. The next day, slides were washed 3x with PBS-T, incubated for 1 h at RT protected from light with secondary antibodies, and washed again. Secondaries included anti-rabbit 594 (Cell Signaling #8889S, Boston, MA, USA) and anti-mouse 488 (ThermoFisher #A-21121) diluted 1:1000 in PBS-T. Slides were then stored in PBS at 4 °C until imaging on a Leica TCS SP8 gated STED microscope.

### 4.5. RNA-Seq Sample and Library Preparation

RMS cell lines (RH4, RH5, or RH41) were plated at 700,000 cells per well into 6-well plates and treated with JQ1 (100 nM), ARV-771 (100 nM), BMS-986158 (50 or 500 nM), or equal volumes of vehicle DMSO (1–15 uL in 1 mL). Cells were treated for 6 h then lysed and homogenized with Qiashredder columns (Qiagen #79654, Hilden, Germany). Bulk RNA was extracted using the RNeasy Plus Mini Kit (Qiagen #74136). Quantity was measured using a Qubit 4 fluorometer and RIN numbers were assessed using a fragment analyzer. Sequencing was performed on Illumina NextSeq 550 High Output Flowcell.

### 4.6. RNA-Seq Analysis

Transcripts per million reads (TPMs) of a gene were measured using RSEM version 1.3.3, which calculated TPM by reassigning multiple alignments of STAR version 2.5.3a to target genes via a maximum likelihood estimation framework. To compare gene set enrichment changes in two samples, we used the Broad Institute and UC San Diego’s GSEA (Gene Set Enrichment Analysis) tool. Custom gene set references were created by selecting up- and downregulated gene sets from the log2-fold change metric, followed by rank ordering. Visualization and summary of GSEA results were performed using custom R scripts (https://github.com/GryderArt/VisualizeRNAseq, last accessed 1 January 2023).

### 4.7. ChIP-Seq

These data were downloaded from the Gene Expression Omnibus (GEO) SuperSeries accession number GSE83728 and visualized in the Integrative Genomics Viewer (IGV) from the Broad Institute.

### 4.8. AQuA-HiChIP Sample Preparation

AQuA-HiChIP was performed as previously described [27] with the Dovetail MNase-HiChIP kit (Dovetail, Catalogue #21007, Sydney, Australia) with a few modifications. For each experiment, 9 × 10^6^ cells were plated into 15 cm plates, treated with a 100 nM concentration of the drug the following morning, and then collected 6 h later and turned into lysates in Stage 1–2. The immunoprecipitation in Stage 3 used approximately 1000 ng of chromatin, 100 ng of C2C12 lysate for spike-in and AQuA normalization, and 10 ug of anti-RNA Pol2 antibody (Santa Cruz Biotechnology #sc-47701 X, Dallas, TX, USA).

### 4.9. HiChIP Analysis

For analysis of 3D loops, the Peaks3D pipeline was used to read HiChIP “.hic” files to make 3D clusters. Filtered loops were made based on the predefined thresholds such as min/max distance restrictions, minimum AQUA-CPM, and the minimum number of connections per cluster. Filtered loops were then merged into a cluster where a path between any ends of the loops was drawn. Briefly, the clustering program began at any loop in the filtered loop set, recursively searched all the loops connected to the left or right ends, and assigned the same cluster ID to the visited ends. This was completed for all unvisited loops with a new cluster ID. The result was 2D peaks, loops (<min_loops), and clusters (≥min_loops), within a given user defined threshold, min_loops. After clustering, these three elements were ranked by the 3D connectivity estimated by the sum of short and long 3D contacts within the expanded boundary (e.g., min and max coordinates of the cluster). This procedure was run iteratively, and the pipeline produced intermediate files such as filtered bedpe, clustered density, QC visualization, and APA plots for user validation.

### 4.10. In Vivo PDX Study

This study protocol was covered by IACUC ACUP CO21-001. Female NOG mice were implanted with CRT00513 tumor fragments intramuscularly into the thigh. When tumors reached an average range of 100–250 mm^3^, mice were randomized to the respective treatment groups, at 7 per group, and were dosed within 24 hrs. The dosing schedule was once daily PO. Tumor volume and bodyweights were measured at least once weekly until growth was observed, and twice weekly after observed growth until study end. Mice were imaged via MRI once before dosing, and once after the last dose. Mice were monitored and dosed for up to 28 days, until animals reached a tumor volume of 1200 mm^3^, or a humane end point, whichever occurred first. Clinical observations were performed twice weekly for any behavioral abnormalities. The length and width of tumors were measured by virtual calipers via MRI and tumor volumes were calculated using the formula 0.5 × L × W^2^.

## 5. Conclusions

We conclude from this study that BRD4 inhibition or degradation is a promising strategy for disrupting the function of PAX3-FOXO1 and the myogenic CR TFs that cooperate with it to drive FP-RMS.

We also conclude that clinical trials for BRD4 inhibitors or degraders should not require MYCN amplification as a criterion for patient inclusion. While in neuroblastoma the presence of an MYCN amplification clearly discriminates tumor subtypes [34], neuroblastoma CR TFs depend on BET bromodomains [35], and MYCN-amplified neuroblastomas respond better to BET bromodomain inhibition [36], we did not see any difference among FP-RMS models with or without MYCN amplification. This is likely because MYCN is a target of PAX3-FOXO1 and plays an important role (and is likely conferring BET bromodomain sensitivity) even when not amplified.

Limitations of this study include: (1) the limited number of animal experiments both in terms of models and the number of BRD4-targeting agents tested head-to-head, and (2) our RNA Pol2 data only reveal the total Pol2 binding and looping, without dissecting changes in the phosphorylation status of Pol2 at enhancers and along the gene body of genes as they are being downregulated. Future studies will also need to explore why there is no defect in pause–release upon BRD4 degradation at CR TF genes, and what other steps of the Pol2 life-cycle may be dependent on BRD4 and its multifaceted biological and biochemical roles.

## Figures and Tables

**Figure 1 pharmaceuticals-16-00199-f001:**
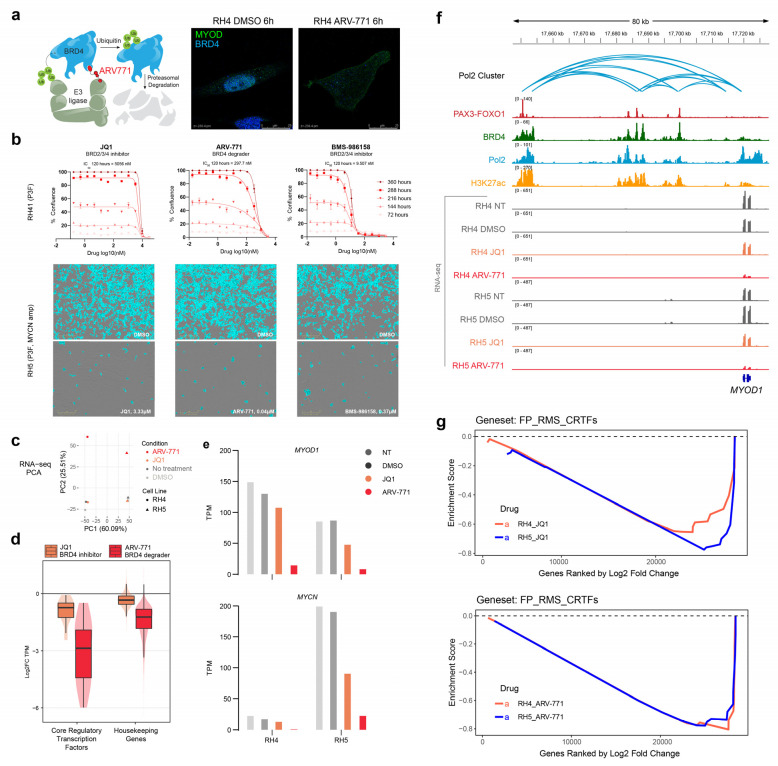
Decrease in cell proliferation and global transcription upon BET bromodomain inhibition and degradation. (**a**) E3-ligase-dependent proteasomal degradation of BRD4 achieved by PROTAC ARV-771 (left), and immunofluorescence of MYOD (green) and BRD4 (blue) in RH4 cells after 6 h of treatment with ARV-771 showing near-complete removal of BRD4 from the cell nucleus. (**b**) Cell proliferation assay of RH41 (P3F+) over 120 h. The BRD4 inhibitor JQ1 has an IC_50_ of 5056 nM; the BRD4 degrader ARV-771 has an IC_50_ 297.7 nM; the BRD4 inhibitor BMS-986165 has an IC_50_ of 9.5 nM. Below are representative images of the effect of BRD4 inhibition or degradation on cell confluence. (**c**) Principal component analysis (top) of RH4 (P3F+) and RH5 (P3F+, *MYCN* amp, mutant p53) treated with DMSO, JQ1, and ARV-771. The first principal component, PC1, shows differences due to cell lines; the second principal component, PC2, shows these differences due to drug treatment. (**d**) Differences in gene expression of core regulatory transcription factors and housekeeping genes across RH4 and RH5 after treatment with the BRD4 inhibitor JQ1 and BRD4 degrader ARV-771. (**e**) Transcripts per million (TPM) of *MYOD1* and *MYCN* in RH4 and RH5 treated with JQ1 or ARV-771. (**f**) Genome browser view of PAX3-FOXO1, BRD4, Pol2, and H3K27ac ChIP-seq data within Pol2 cluster at *MYOD1*, overlaid with RNA-seq expression tracks in RH4 and RH5 treated with JQ1 or ARV-771. Pol2 HiChIP loops are plotted as arcs. (**g**) Gene set enrichment analysis (GSEA) of RH4 and RH5 cells treated with JQ1 and ARV-771 shows strong downregulation of FP-RMS CRTFs.

**Figure 2 pharmaceuticals-16-00199-f002:**
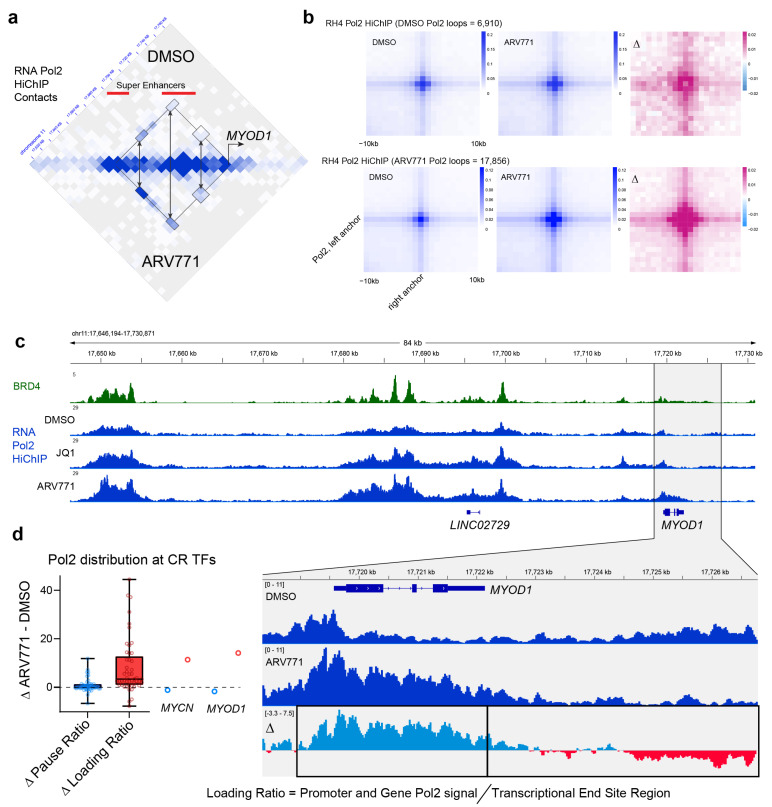
Pol2 redistribution upon the loss of BET bromodomain proteins. (**a**) Contact matrix of RNA Pol2 AQuA-HiChIP in RH4 treated with DMSO (top) or ARV-771 (bottom). H3K27ac defined super enhancers are annotated with red lines. Color scales from gray to blue indicate the number of contacts connecting distal genomic sites. (**b**) Aggregate peak analysis of Pol2 loops called in DMSO- (top) or ARV-771-treated RH4 cells (bottom). Scales are in contacts per million per loop and are AQuA-normalized. (**c**) Genome browser tracks for BRD4 ChIP-seq (top) and RNA Pol2 HiChIP in RH4 treated with DMSO, JQ1, or ARV-771. (**d**) RNA Pol2 distribution changes upon ARV-771 treatment, measured by changes in the pause ratio or loading ratio. Data plotted are from Pol2 HiChIP at FP-RMS CR TFs (core regulatory transcription factors).

**Figure 3 pharmaceuticals-16-00199-f003:**
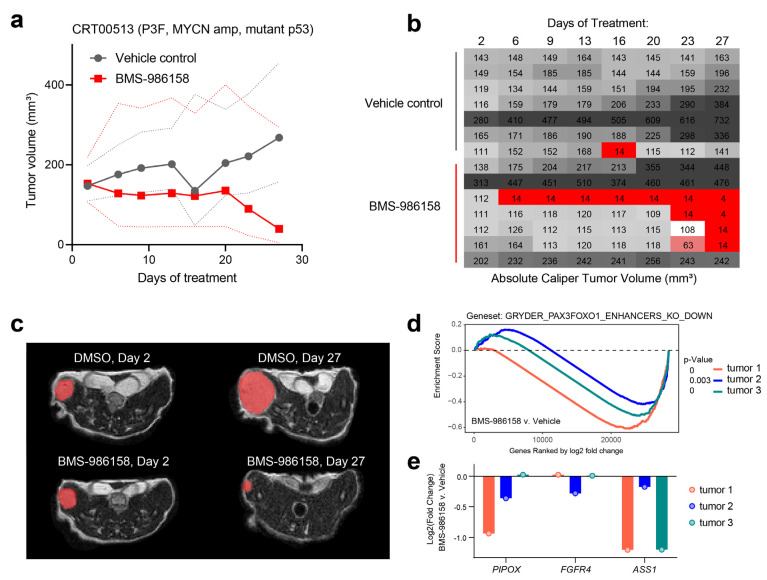
Effect of potent BRD4 inhibitor on FP-RMS O-PDX tumor growth in vivo. (**a**) Mice were implanted with FP-RMS PDX CRT00513 intramuscularly in the thigh and, after tumors reached a threshold volume, were dosed with vehicle or BMS-986158 once daily for 28 days. Average FP-RMS O-PDX tumor volume, measured by digital calipers via MRI, over time with vehicle or BMS-986158 treatment (average = solid line, 95% confidence interval = dashed line). (**b**) Tumor volumes for individual mice within each condition at each time point. (**c**) Demonstrative axial section of T2-weighted MRIs at days 2 and 27 of DMSO or BMS-986158-treated mice. Tumors highlighted in red. (**d**) Gene set enrichment analysis (GSEA) of RNA-seq from CRT00513 tumors treated with BMS-986158 for 3 days in vivo shows strong downregulation of PAX3-FOXO1 target genes. (**e**) Variability in gene expression responses for PAX3-FOXO1 target genes *PIPOX*, *FGFR4* and *ASS1* across the BMS-986158-treated tumors.

## Data Availability

The data presented in this study will be uploaded to GEO.

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
