# Peer review of "BET Bromodomain Degradation Disrupts Function but Not 3D Formation of RNA Pol2 Clusters"

_pharmaceuticals, 2023, doi:10.3390/ph16020199_

Round 1

Reviewer 1 Report

1.  In P.2 lines 85-7, the statement “This suggests that both degradation and inhibition can have potent effects on cell growth, but that degradation is more potent than inhibition for a given BRD4-binding small molecule” needs to be revised because the IC50 of BMS-986158 (9.507 nM), an inhibitor was much (~30-fold) lower than that of ARV771 (297.7 nM), a degrader.

2.   The results of the animal study were very puzzling because 3 mice responding to BMS-986158 treatment showed (only) a sudden decrease of their tumors at day 23, 27, and 23 & 27, respectively, and the size of tumor in 1 non-responder remained at ~ 240 mm3 (Fig. 3b). Moreover, they could use tumor growth curves to replace the table which might be easier to understand. As pointed out by the authors, a larger study with BMS-986158 with greater statistical power is required to assess whether this compound really suppresses tumor growth.

3.   They should try to find out why some PDXs did not respond (or respond) to BMS-986158 treatment by performing IHC staining of the tumor tissues harvested after sacrificing the animals.

Author Response

1. In P.2 lines 85-7, the statement “This suggests that both degradation and inhibition can have potent effects on cell growth, but that degradation is more potent than inhibition for a given BRD4-binding small molecule” needs to be revised because the IC50 of BMS-986158 (9.507 nM), an inhibitor was much (~30-fold) lower than that of ARV771 (297.7 nM), a degrader.

We have amended to correct this, and now the text reads: "Comparing the effects of JQ1 and ARV-771, which have similar BET-binding moieties, suggests that degradation is more potent than inhibition for a given BRD4 binding moiety, while the superior potency of BMS-986158 compared to ARV-771 shows that inhibition can outperform degradation with an improved BRD4 binding moiety"

2. The results of the animal study were very puzzling because 3 mice responding to BMS-986158 treatment showed (only) a sudden decrease of their tumors at day 23, 27, and 23 & 27, respectively, and the size of tumor in 1 non-responder remained at ~ 240 mm3 (Fig. 3b). Moreover, they could use tumor growth curves to replace the table which might be easier to understand. As pointed out by the authors, a larger study with BMS-986158 with greater statistical power is required to assess whether this compound really suppresses tumor growth.

Yes, we agree this is very puzzling.  In a future study, we plan on performing a larger cohort, and also plan to include other PDX models for FP-RMS, to try and see if this is a real phenomenon. 

3. They should try to find out why some PDXs did not respond (or respond) to BMS-986158 treatment by performing IHC staining of the tumor tissues harvested after sacrificing the animals.

This was an important point.  To attempt to resolve these issues, we did a short (3 day) dosing of the drug (or vehicle) in 3 mice bearing this same tumor model as the efficacy study, followed by RNA-seq.  We did find variable strength of gene expression responses, but all tumors showed a decrease in PAX3-FOXO1 target gene expression (new Figure 3d).  Unfortunately, we do not have tumors from the longer-term efficacy to perform IHC staining, which would have been ideal.

Reviewer 2 Report

This manuscript entitled "BET bromodomain degradation disrupts function but not 3D formation of RNA Pol2 clusters" by Chin et al. indicated roles of BRD4 including BET bromodomain to  FP-RMS via transcriptional regulation. This study is very important and inetresting. But some corrections may be needed. It should be added "Conclusion Section". In addition, it is better to add more detail about molecular mechanisms of FP-RMS to pathgenesis and  myogenic differentiation via small non-coding RNAs and histon deacethylase etc. Also, it  is better to add advantages/disadvantages of this study in Discussion section. 

Author Response

This manuscript entitled "BET bromodomain degradation disrupts function but not 3D formation of RNA Pol2 clusters" by Chin et al. indicated roles of BRD4 including BET bromodomain to  FP-RMS via transcriptional regulation. This study is very important and inetresting. But some corrections may be needed. It should be added "Conclusion Section". In addition, it is better to add more detail about molecular mechanisms of FP-RMS to pathgenesis and  myogenic differentiation via small non-coding RNAs and histon deacethylase etc. Also, it  is better to add advantages/disadvantages of this study in Discussion section. 

Thank you for the supportive review.  We have now (1) added a Conclusion Section, (2) added more detail about the molecular mechanisms of FP-RMS, and including new references to support this, and (3) we have added statements about the limitations of the study to the Conclusions section.